# ON LOTTERY TICKETS AND MINIMAL TASK REPRESENTATIONS IN DEEP REINFORCEMENT LEARNING

**Marc Vischer** *
Technical University Berlin

Robert Tjarko Lange *
Technical University Berlin
Science of Intelligence

Henning Sprekeler
Technical University Berlin
Science of Intelligence

## ABSTRACT

The lottery ticket hypothesis questions the role of overparameterization in supervised deep learning. But how is the performance of winning lottery tickets affected by the distributional shift inherent to reinforcement learning problems? In this work, we address this question by comparing sparse agents who have to address the non-stationarity of the exploration-exploitation problem with supervised agents trained to imitate an expert. We show that feed-forward networks trained with behavioural cloning compared to reinforcement learning can be pruned to higher levels of sparsity without performance degradation. This suggests that in order to solve the RL problem agents require more degrees of freedom. Using a set of carefully designed baseline conditions, we find that the majority of the lottery ticket effect in both learning paradigms can be attributed to the identified mask rather than the weight initialization. The input layer mask selectively prunes entire input dimensions that turn out to be irrelevant for the task at hand. At a moderate level of sparsity the mask identified by iterative magnitude pruning yields minimal task-relevant representations, i.e., an interpretable inductive bias. Finally, we propose a simple initialization rescaling which promotes the robust identification of sparse task representations in low-dimensional control tasks.

## 1 INTRODUCTION

Recent research on the lottery ticket hypothesis (LTH, Frankle & Carbin, 2019; Frankle et al., 2019) in deep learning has demonstrated the existence of very sparse neural networks that train to performance levels comparable to those of their dense counterparts. These results challenge the role of overparameterization in supervised learning and provide a new perspective on the emergence of stable learning dynamics (Frankle et al., 2020a;b). Recently these results have been extended to various domains beyond supervised image classification. These include self-supervised learning (Chen et al., 2020a), natural language processing (Yu et al., 2019; Chen et al., 2020b) and semantic segmentation (Girish et al., 2020). But how does the lottery ticket ticket phenomenon transfer to reinforcement learning agents? One key challenge may be the inherent non-stationarity of the optimization problem in deep reinforcement learning (DRL): The data-generation process is not static, but depends on the changing state of the neural network. Furthermore, a weight may serve different roles at different stages of learning (e.g. during exploration and exploitation). It is not obvious how a simple weight magnitude-based pruning heuristic acting on a well-performing policy shapes the learning process of the agent. In this work, we therefore investigate winning tickets in reinforcement learning and their underlying contributing factors. We compare supervised behavioral cloning with DRL, putting a special emphasis on the resulting input representations used for prediction and control. Thereby, we connect the statistical perspective of sparse structure discovery (e.g. Hastie et al., 2019) with the iterative magnitude pruning (IMP, Han et al., 2015) procedure in the context of Markov decision processes (MDPs). The contributions of this work are summarized as follows:

1. We show that winning tickets exist in both high-dimensional visual and control tasks (continuous/discrete). A positive lottery ticket effect is robustly observed for both off-policy DRL algorithms, including Deep-Q-Networks (DQN, Mnih et al., 2015) and on-policy

---

*Authors contributed equally. RTL is the corresponding author (`robert.t.lange@tu-berlin.de`).

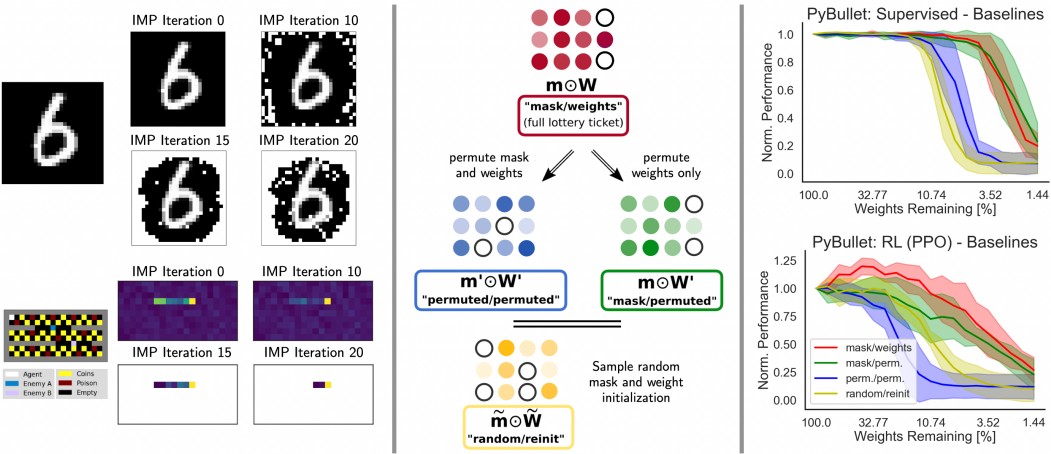

Figure 1: Representation compression, disentangling baselines & lottery ticket effects in continuous control tasks. **Left.** IMP successively prunes task-irrelevant outer rim pixels in a MNIST digit-classification task and for an IMP-masked agent solving a visual navigation task in DRL. The channel encoding the patrolling enemy is pruned up to the point where only potential enemy locations are considered. **Middle.** To disentangle the contributions of mask, initialization and layer-wise pruning ratio to the winning lottery ticket (*mask/weights* condition), we compare three baselines: After each IMP iteration, we permute either only the remaining initial weights (*mask/permuted*) or also the sparsity mask (*permuted/permuted*). The third baseline is created by randomly sampling a sparse mask and random re-initialization of the weights (*random/re-init*). **Right.** Avg. normalized performance of policies at different sparsity levels and for different baseline conditions and four PyBullet (Ellenberger, 2018) tasks. For both behavioral cloning and PPO agents most of the ticket effect can be attributed to tasks IMP-derived mask as compared to the weight initialization. Agents trained with supervision can be pruned to higher sparsity levels before performance deteriorates.

policy-gradient methods (PPO, Schulman et al., 2015; 2017), providing evidence that the lottery ticket effect is a universal phenomenon across optimization formulations in DRL.

2. By comparing RL to supervised behavioral cloning (BC), we show that networks trained with explicit supervision can be pruned to higher sparsity levels before performance starts to degrade, indicating that the RL problem requires a larger amount of parameters to address exploration, distribution shift & quality of the credit assignment signal (section 3.1).

3. By introducing a set of lottery ticket baselines (section 3; figure 1), we disentangle the contributions of the mask, weight initialization and layer-wise pruning ratio. We demonstrate that the mask explains most of the ticket effect in behavioral cloning and reinforcement learning for MLP-based agents, whereas the associated weight initialization is less important (section 3.2 and 3.3). For CNN-based agents the weight initialization contributes more.

4. By visualizing the sparsified weights for each layer, we find that early network layers are pruned more. Entire input dimensions can be rendered invisible to MLP-based agents by the pruning procedure. By this mechanism, IMP compresses the input representation of the MDP (e.g. figure 1, left column, bottom row) and reveals a minimal task representation for the underlying control problems (section 4).

5. The IMP input layer mask not only eliminates obviously redundant dimensions, but also identifies complex relationships between input features and the task of the agent (section 4.1), e.g. the proximity of an enemy or the speed of an approaching object. We show that the input masking can be transferred to train dense agents with sparse inputs at lower costs.

6. We show that the weight initialization scheme is important for discovering minimal representations. Depending on the input size of different layers of the network, global magnitude-based pruning can introduce a strong layer-specific pruning bias. We compare initializations and show that a suitable initialization scheme enables the removal of task-irrelevant dimensions (section 4.2).

## 2 BACKGROUND AND RELATED WORK

**Iterative Magnitude Pruning.** We use the iterative pruning procedure outlined in Frankle & Carbin (2019) to identify winning tickets. We train DRL agents for a previously calibrated number of transitions and track the best performing network checkpoint throughout. Performance is measured by the average return on a set of evaluation episodes. Afterwards, we prune $20\%$ of the weights with smallest magnitude globally (across all layers). The remaining weights are reset to their initial values and we iterate this procedure (train $\rightarrow$ prune $\rightarrow$ reset).[1] The *lottery ticket effect* refers to the performance gap between the sparse network obtained via IMP and a randomly initialized network with sparsity-matched random pruning mask.

**Lottery Tickets in Deep Reinforcement Learning.** Yu et al. (2019) previously demonstrated the existence of tickets in DRL that outperform parameter-matched random initializations. They obtained tickets for a distributed on-policy actor-critic agent on a subset of environments in the ALE benchmark (Bellemare et al., 2013) as well as a set of discrete control tasks. While they provide empirical evidence for the existence of lottery tickets in DRL, they did not investigate the underlying mechanisms. Here, we aim to unravel these mechanisms. To this end, we focus on a diverse set of environments and provide a detailed comparison between supervised behavioral cloning and on-/off-policy Deep RL with a set of carefully designed ticket baselines. We analyze the resulting masked representations that the agent learns to act upon and the impact of specific weight initializations on the resulting sparse networks.

**Lottery Tickets with Non-Stationary Data Distributions.** Desai et al. (2019) investigated whether trained lottery tickets overfit the training data distribution under which they were obtained. Using transfer learning tasks on natural language data, they showed that lottery tickets provide general inductive biases. Similar ticket transfer results were reported by Morcos et al. (2019) and Mehta (2019) in the context of optimizers and vision datasets. Unlike our work, these studies do not investigate within-training covariate shift, but instead focus on transferring ticket initializations after a full IMP run. Chen et al. (2021), on the other hand, investigate the ticket phenomenon in the context of lifelong learning and class-incremental image classification. They propose new pruning strategies to overcome the sequential nature of tasks and need for increased model capacity. Compared to the DRL setting, the covariate shift is here determined by the curriculum schedule of tasks and not the exploration behaviour of the network-parameterized agent.

**Deep Reinforcement Learning Background.** In our off-policy DRL experiments, we train Deep-Q-Networks (DQN, Mnih et al., 2015) with double Q-learning loss (Van Hasselt et al., 2016) and prioritized experience replay (Schaul et al., 2015). As a representative on-policy algorithm, we chose Proximal Policy Optimization (PPO, Schulman et al., 2015; 2017). PPO is a baseline-corrected policy gradient algorithm which uses a clipping strategy to approximate a computationally expensive trust-region optimization method. For illustrative purposes, we train DQN agents on a visual navigation task, in which an agent has to collect coins in a grid while avoiding poison and two patrollers that are moving in restricted parts of the grid (figure 1, left column, bottom row; SI B). We scale our results to four PyBullet (Ellenberger, 2018) continuous control and a subset of ALE benchmark (Bellemare et al., 2013) environments. Due to computational considerations we limit each individual IMP iteration for the ATARI environments to 2.5 million frames. All other tasks were trained for a pre-calibrated generous amount of transitions. We focus on feedforward value estimators and policies (MLP & CNN) and used default hyperparameters with little tuning (SI C).

**Supervised Behavioral Cloning.** While most supervised learning relies on a stationary data distribution provided by a static dataset, reinforcement learning agents have to acquire their training data in an action-perception loop. Since the agent's behavioural policy is learned over time, the data distribution used in optimization undergoes covariate shift. To study how the covariate shift, additional exploration problem and different credit assignment signal influence winning tickets, we mimic the supervised learning case by training agents via supervised policy distillation (Rusu et al., 2015; Schmitt et al., 2018). We roll out a pre-trained expert policy and train the student agent by minimizing the KL divergence between the student's and teacher's policies.

---

[1]In supervised learning, the pruning mask is often constructed based on an early stopping criterion and the final network. We instead track the best performing agent. Thereby, we reduce noise introduced by unstable learning dynamics and exploit that the agent is trained and evaluated on the same environment. We found that late rewinding to a later checkpoint (Frankle et al., 2019) is not necessary for obtaining tickets (SI figure 14).

# 3   DISENTANGLING TICKET CONTRIBUTIONS IN BC AND DEEP RL

There are two contributing factors to the lottery ticket effect: The IMP-identified binary mask and the preserved initialized weights that remain after pruning (*mask/weights*). We aim to disentangle the contributions by introducing a set of counterfactual baselines, which modify the original IMP procedure (figure 1, middle column; table 1). A first baseline estimates how

| Examined Sparsity-Generating IMP Variants | | | |
|---|---|---|---|
| | Retain weights | Retain mask | Retain layer pruning ratio |
| *mask/weights* | ✓ | ✓ | ✓ |
| *mask/permuted* | ✗ | ✓ | ✓ |
| *permuted/permuted* | ✗ | ✗ | ✓ |
| *random/re-init* | ✗ | ✗ | ✗ |

Table 1:   Baselines for Disentangling Ticket Contributions

much of the performance of the ticket can be attributed to the initial weights, by means of a layer-specific permutation of the weights that remain after masking (*mask/permuted*). A second, weaker baseline estimates the contribution of the mask, by also permuting the layer-specific masks (*permuted/permuted*). Finally, we consider the standard *random/re-init* baseline, which samples random binary masks – discarding layer-specific pruning ratios – and re-initializes all weights at each IMP iteration. Throughout the next sections we use these baselines to analyze and compare the factors that give rise to the lottery ticket effect in different control settings.

## 3.1   COMPARING WINNING TICKETS IN SUPERVISED BEHAVIORAL CLONING AND DEEP RL

Does the covariate shift in DRL affect the existence and nature of lottery tickets? Weights pruned for their small magnitude at the end of the learning process might be needed at earlier stages, e.g., during exploration. If this were the case, the performance of the ticket in DRL should degrade for lower levels of sparsity than in a corresponding supervised task. To investigate this question, we turn to a behavioral cloning setting, in which a student agent is trained to imitate the stochastic policy of a pre-trained expert and apply magnitude pruning to the student network after each training run iteration. By collecting transitions based on the static behavioral policy of the expert, we avoid the need for exploration and the effect of an otherwise non-stationary data distribution. For both the discrete action space and all continuous control tasks we find that agents trained with RL and the supervised settings start to degrade in performance at different sparsity levels (figure 2). More specifically, the maximal return of agents that were trained with reinforcement learning starts to drop at significantly earlier stages of the IMP procedure. Hence, the covariate shift inherent to the RL problem increases the minimal required size of a winning ticket in the tasks studied here: More parameters are required for an agent that has to solve the additional exploration problem.

## 3.2   DISENTANGLING TICKET CONTRIBUTIONS IN SUPERVISED BEHAVIORAL CLONING

To disentangle the contributions of the initial weights and the weight mask to the ticket effect in supervised behavioral cloning, we next compared the three baselines to the full non-randomized ticket (figure 1, top, right and 3). For most agents, training performance does not degrade substantially when the weights are permuted but the mask is kept intact (no significant gap between red and green curves). But we observe a strong decrease in the ability to prune the policy to high sparsity levels if one additionally permutes the IMP-derived weight mask (gap between green and blue curves). This observation holds for the cloning of expert policies in both discrete and continuous control tasks. Hence, the ticket effect can be mainly attributed to the mask rather than the weights. Finally, the traditional *random/re-init* baseline performs worse already for moderate levels of sparsity. The resulting performance gap (blue and yellow curves) indicates the contribution of the layer-wise pruning ratio to the mask effect. These insights emphasize the importance of strong and nuanced baselines to understand the contributions to the full lottery ticket effect.

## 3.3   DISENTANGLING TICKET CONTRIBUTIONS IN DEEP REINFORCEMENT LEARNING

The observation that ticket information is mostly carried by the mask rather than the initial weights carries over to the full DRL setups (figure 1, bottom, right and figure 4).

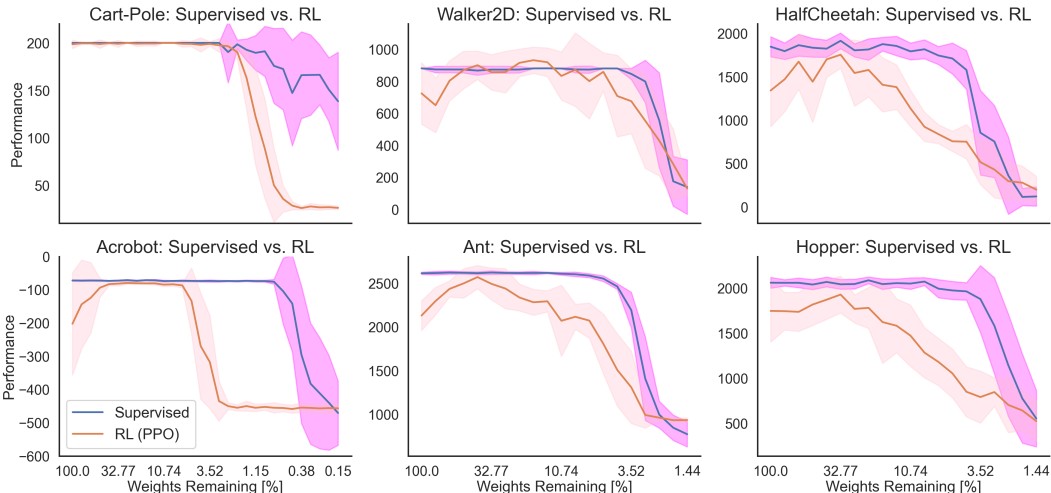

Figure 2: Comparing lottery tickets in DRL and supervised behavioral cloning. Networks trained with explicit supervision can be pruned to higher sparsity levels before performance starts to degrade. Results are averaged over 15 runs for both the Cart-Pole and Acrobot and 10 runs for PyBullet environments. We plot mean best performance and one standard deviation.

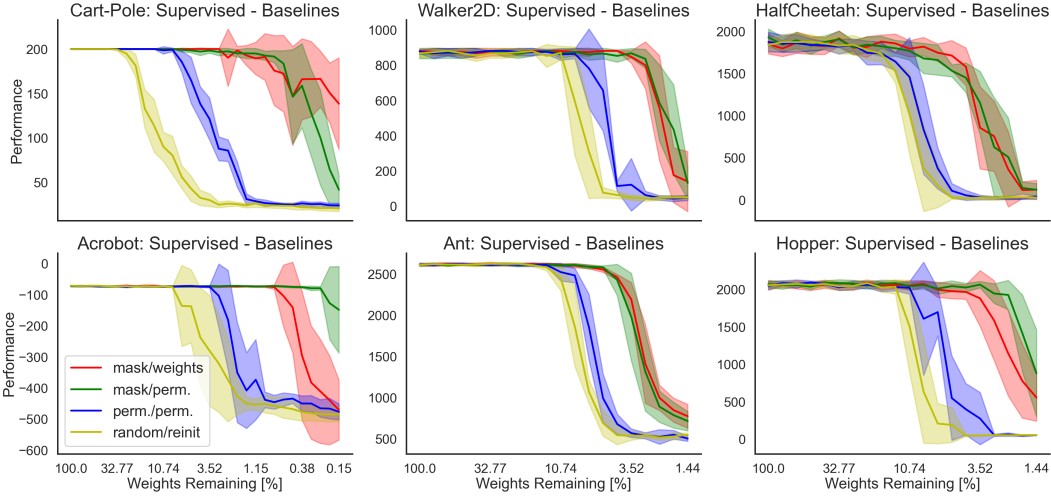

Figure 3: Disentangling baselines for lottery tickets in supervised behavioral cloning. The gap between the ticket (*mask/weights*) and weight-permuted baseline (*mask/permuted*) is small, indicating a strong contribution of the mask. Results are averaged over 15 runs for both Cart-Pole and Acrobot and 10 runs for PyBullet environments. We plot mean best performance and one standard deviation.

When preserving the information contained in the mask, agents can be pruned to much higher levels of sparsity as compared to the case in which the mask information is distorted (permuted or resampled). This effect holds for both PPO- and DQN-based agents and across a wide range of qualitatively different tasks.[2] Interestingly, ticket weights are more important for CNN-based agents (e.g., see ATARI plots). We provide further experimental evidence for this observation on four MinAtar games (Young & Tian, 2019) in the supplementary material (SI figure 10), where we compare MLP- and CNN-based agents.

Unlike the behavioral cloning experiments, for some tasks the *permuted/permuted* baseline performs worse than the *random/reinit* configuration. This observation only occurs for the PyBullet control

---

[2]For further experiments investigating the role of different network capacities (SI figures 11, 12), state encoding (SI figure 13) and regularization (SI figure 14) we refer the interested reader to the SI.

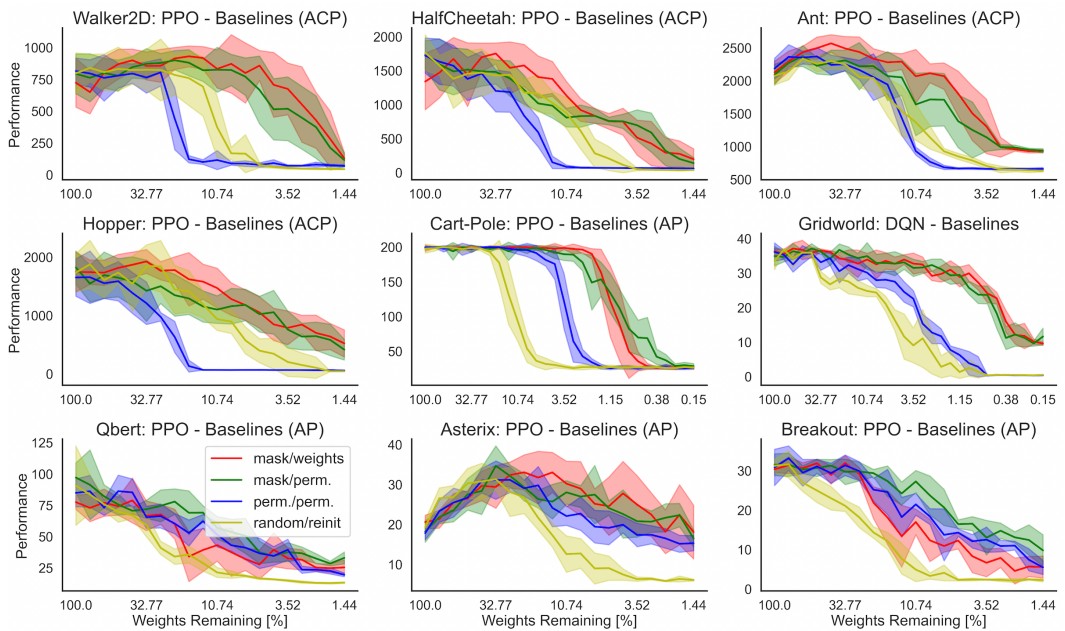

Figure 4: Tickets in on- and off-policy deep reinforcement learning. The disentangling baselines for tickets in on-policy (PPO) and off-policy (DQN) DRL for a set of continuous control, a visual navigation and a subset of ATARI environments reveal the consistent importance of the IMP-extracted mask. Results are averaged over 5 independent runs on the Gridworld and ATARI and 10 runs on the continuous control environments. We plot mean best performance and one standard deviation.

tasks in which both the separate critic and actor networks were pruned (ACP). If instead, one only prunes the actor network (AP), as was done for the PPO agents trained on Cart-Pole, Acrobot and ATARI, the *permuted/permuted* baseline performs stronger (see SI figure 15). This result highlights the importance of differences in IMP-derived pruning ratios between agent modules, i.e. separate actor and critic networks.

## 4    MINIMALLY TASK-SUFFICIENT REPRESENTATIONS VIA IMP

The previous results establish the importance of the information contained in the mask for the success of winning tickets. To better understand this phenomenon and the resulting inductive biases, we analyzed the sparsified weight matrices, specifically between the input and the units in layer one. The next section sets out to answer the following questions: **1)** The derived mask can be thought of as a pair of goggles which guide the processing of state information. What do the masked MLP-based agents observe? **2)** Can we transfer the input layer mask and re-use it as an inductive bias for otherwise dense agents? **3)** How much influence do weight initialization and input dimensionality have on the discovered input layer mask?

### 4.1    MINIMAL TASK REPRESENTATIONS IN HIGH-DIMENSIONAL VISUAL TASKS

In the visual navigation experiments, the pruning primarily affects the input layer (figure 5, top right column). When visualizing the cumulative absolute weights of the input layer for an IMP-derived DQN agent, we find that IMP deletes entire dimensions from the observation vector (figure 5, left column) by removing all connections between those dimensions and the first hidden layer. These eliminated input dimensions are not task-relevant. A fully-connected network trained on the remaining dimensions learns the task as quickly as when trained on the complete set of input dimensions (figure 5, bottom right column). Hence, the IMP-derived mask provides a compressed representation of a high-dimensional observation space, enabling successful training even at high levels of sparsity.

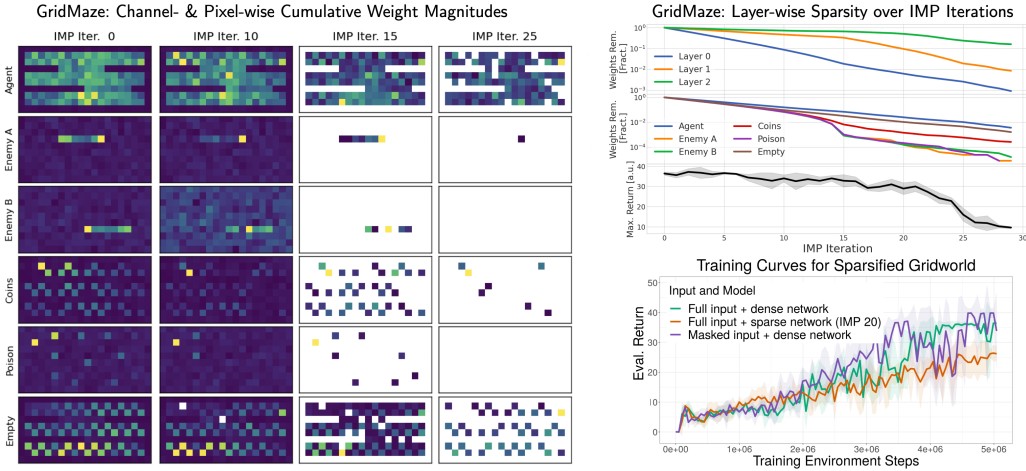

Figure 5: IMP eliminates task-irrelevant observation dimensions for a high-dimensional visual navigation task ($o_t \in \mathbb{R}^{6 \times 10 \times 20}$). **Left.** Channel-/pixel-wise cumulative weight magnitudes. IMP successively prunes redundant input pixels which are not necessary to solve the navigation task. All of the pruned enemy channel pixels encode locations which the patrolling enemy cannot access. **Right, Top.** Pruning affects the layers and object channels differentially. The input layer is pruned most strongly, while the agent channel is pruned the least. **Right, Bottom.** The input layer pruning mask (at moderate sparsity levels) can be used as an inductive bias for training a dense network.

This result extends to masked input layers of MLP-agents trained on the MinAtar environments (figure 6 and SI). For the Freeway task we find that the IMP mask encodes the notion of velocity of moving cars. The input layer weights corresponding to the binary observation channel which encodes the slowest moving car are pruned the most, maintaining only the pixels needed to react in time. The same principle applies to the enemy bullet channel in the SpaceInvaders task. The agent only needs to know about a potential collision at the next time step in order to avoid being hit. Therefore, IMP prunes all information about the enemy bullet that is more than one step away from the agent. For all visualized MinAtar tasks, only the actionable row/column of the agent channel is preserved. In Breakout, additionally the bottom row of the ball and trail channel is pruned since the game terminates once the ball or trail reach it. There is no action-relevant information encoded and it can be discarded. In summary, we find that IMP-derived input layer mask provides a visually interpretable and physically meaningful compression of the observation space.

## 4.2 MINIMAL REPRESENTATIONS IN LOW-DIMENSIONAL CONTROL TASKS AND THE ISSUE OF INITIALIZATION SCALES

The continuous and discrete control tasks, on the other hand, rely on low-dimensional state representations. At first sight, they do not contain obviously uninformative input dimensions that lend themselves to be pruned. We would expect the cumulative weight strength of each input dimension to decrease at equal speed over the course of iterative pruning. Contrary to this initial intuition, IMP still aggressively prunes core dimensions while yielding trainable agents (figure 7, 8). For a subset of continuous control tasks, we find that one can prune 20 and up to 50 percent of the input dimensions, while still being able to train the agent to the performance level of a dense counterpart (figure 7). The exact amount depends on the considered environment indicating a varying degree of observation over-specification across task formulations. For example, the Walker2D environment can train to full performance with only 11 out of 22 observation dimensions, while the Ant environment requires 22 out of 28 dimensions. We note an initial positive effect of the weight pruning on the performance of all agents indicating the effectiveness of the implied regularization.

In the Cart-Pole task the cart position and its velocity are pruned (figure 8, top). Newton's laws are invariant with regard to the choice of the inertial frame of reference, so the input channels encoding the cart position and velocity do not provide additional information over the pole's angle and velocity. The Acrobot task requires the agent to swing-up a two-link pendulum across a line as fast

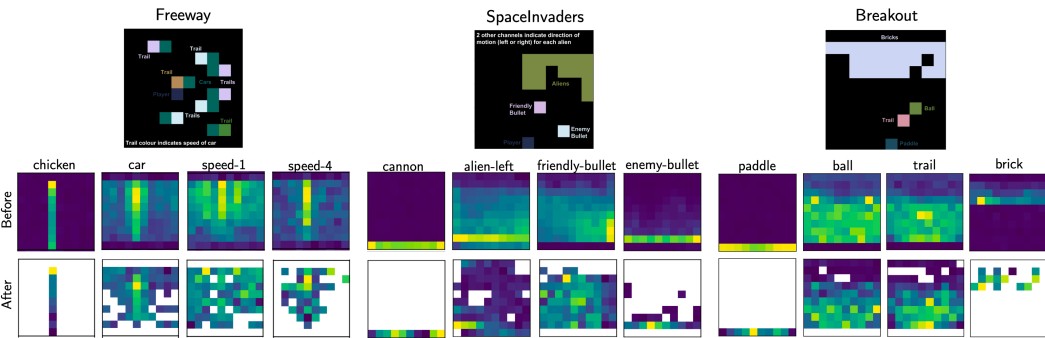

Figure 6: IMP eliminates task-irrelevant observation dimensions for a selection of MinAtar environments (Young & Tian, 2019). The environment depicting figures were adapted from Young & Tian (2019). **Left.** Freeway. IMP provides an inductive bias by differentially pruning object channels with different velocities (e.g. car speeds). **Middle.** SpaceInvaders. IMP only preserves pixels of enemy objects which encode actionable proximity information (e.g. bullets being close to the agent). **Right.** Breakout. IMP discards pixels which only change when it is too late to act (e.g. when the ball has left the display and the episode terminates).

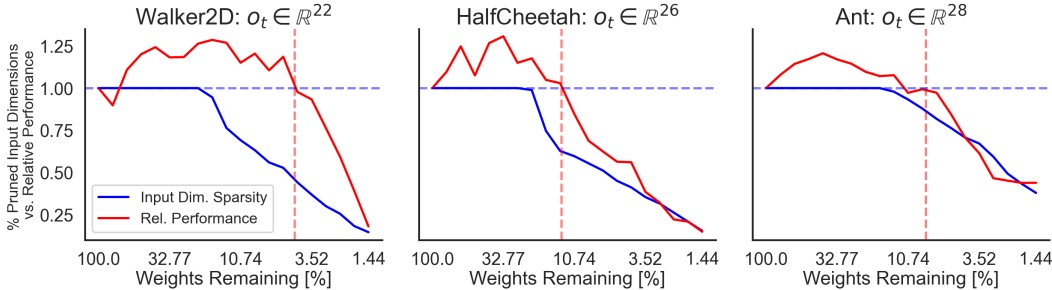

Figure 7: IMP eliminates task-irrelevant observation dimensions for a set of continuous control tasks. For all three tasks there exists a moderate sparsity level (approximately 10% non-sparse weights) at which entire input units (columns of the first linear layer) are pruned while the agents still train to the performance level of their dense counterparts. The extent of this observation strongly depends on the considered environment indicating a varying extent of observation over-specification.

as possible. Here, IMP eliminates all sine and cosine transformations of the rotational angles of the links (figure 8, middle row, right column). Since the swing-up can be achieved by coordinating and increasing the angular velocities of the two links, only these two dimensions have to be preserved from pruning. These experiments demonstrate that IMP can yield fundamental insights into what physical information is sufficient to solve a task and leverages them to prune the input representation without impairing performance. During our analysis we discovered that the core dimension discovery of IMP crucially depends on the initialization of the input layer weights. Popular weight initialization heuristics such as the Kaiming family (He et al., 2015) aim to preserve activation magnitudes across network layers. As a result, layers that receive high-dimensional input are initialized at lower values and are therefore prone to more aggressive pruning by IMP. This bias does not occur for Xavier initializations (Glorot & Bengio, 2010, figure 8, top rows, right column). Furthermore, we experimented with different approaches to combat this initialization bias. We find that downscaling the Kaiming initialization of the input layer by a factor of ten leads to a robust discovery after few IMP iterations (figure 8, top rows, middle column). While smaller rescaling factors lead to a slower separation of dimensions, larger factors result in overly aggressive pruning of the input layer. In summary, we have shown that the input layer initialization can shape the resulting lottery ticket mask and provide a first initialization heuristic promoting efficient minimal task representation discovery.

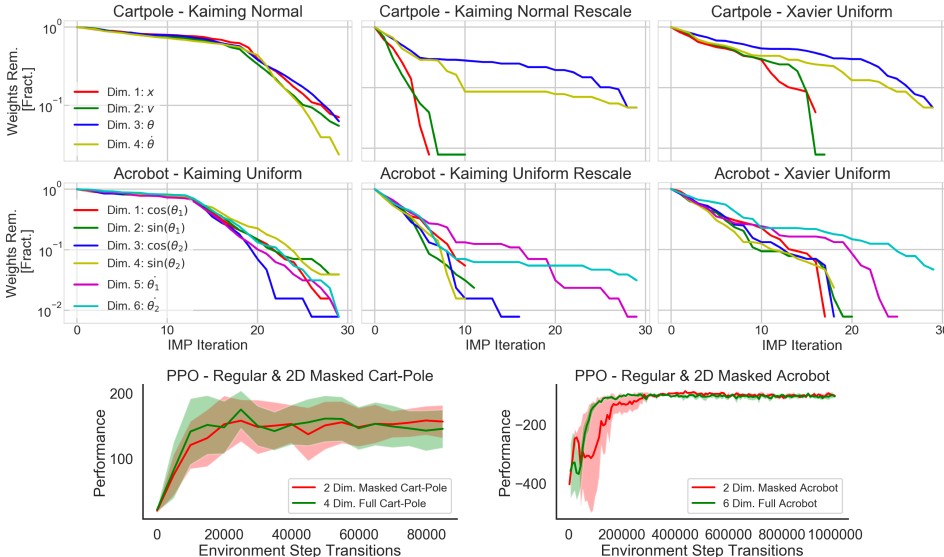

Figure 8: IMP eliminates task-irrelevant observation dimensions for low-dimensional control tasks. Suitable initialization heuristics can promote and accelerate the discovery of minimal representations. **Top.** IMP identifies that the Cart-Pole task can be solved with only two dimensions: The pole angle and angular velocity. **Middle.** IMP identifies that the Acrobot task can be solved with only two dimensions: The angular velocities of the two pendulum links. **Bottom.** Agents can successfully be trained on the subset of IMP-derived dimensions alone. We plot mean performance and one standard deviation across 10 independent runs.

## 5 CONCLUSION

**Summary.** This work has investigated the mechanisms underlying the lottery ticket effect in behavioral cloning and DRL. We provide evidence that agents which are trained on a stationary data distribution and do not face the exploration problem, can be pruned to higher levels of sparsity while successfully performing their task. For both setups we find that the effect can be attributed to the identified mask rather than to the corresponding weight initialization. We have shown that the mask compresses the input representation by removing task-irrelevant information in the form of visual object pixels or entire control observation dimensions. Finally, we found that the pruning process is sensitive to the layer-specific scale of the weight initialization. The interpretability of the resulting minimal representations can be enhanced by heuristically re-scaling the weights in different layers.

**Limitations.** While our observations are qualitatively robust, they quantitatively vary across learning paradigms, algorithms, architectures and different tasks. Compared to our results, weight values have been claimed to play a more central role in the context of lottery tickets in supervised image classification (Frankle et al., 2020a). Future work will need to investigate under which conditions the initialization of the weight values significantly contributes to the ticket effect in RL. Furthermore, most of our experimental results have been obtained for on-policy PPO and policy-based BC agents. Whether the interpretation of the mask effect as a representation regularizer also holds for intermediate and higher layer masks is hard to assess and remains an open question. Finally, this study is empirical in its nature and will require further theoretical guidance and foundations.

**Future Work.** In future work we want to investigate layer-wise effects, e.g., by studying layer-specific pruning ratios, normalization schemes and fine-grained baseline analyses, in the hope of identifying sparser, more efficient and interpretable winning tickets. Further work needs to be done to disentangle the contributions of the additional exploration problem, distribution shift and credit assignment signal in RL. Our proposed baseline analysis can be applied more generally to all studies considering the lottery ticket procedure. Finally, we believe that there are many opportunities to modify the original IMP procedure to simplify the discovery of minimal representations. For example, pruning entire units of intermediate layers may improve the interpretability of hidden representations.

ACKNOWLEDGMENTS

We thank Jonathan Frankle for initial discussions and feedback on the first manuscript draft. Furthermore, we thank Joram Keijser for reviewing the manuscript. This work is funded by the Deutsche Forschungsgemeinschaft (DFG, German Research Foundation) under Germanys Excellence Strategy - EXC 2002/1 Science of Intelligence - project number 390523135.

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

SUPPLEMENTARY INFORMATION: ON LOTTERY TICKETS AND MINIMAL
TASK REPRESENTATIONS IN DEEP REINFORCEMENT LEARNING

## A    ADDITIONAL RESULTS

### A.1    MINATAR DISTILLATION AND DQN RESULTS - MLP AND CNN-BASED AGENTS

To test the robustness of the lottery ticket phenomenon to different architectures and diverse tasks, we repeat the baseline comparison distillation experiments for the MinAtar environments (see figure 9). We trained MLP- and CNN-based agents to distill experts' value estimators and using the same architecture and hyperparameters for all considered games. For MLP agents the mask consistently contributes most to the the ticket. For CNN-based agents and selected games (Asterix and Space Invaders) the weight initialization contributes more.

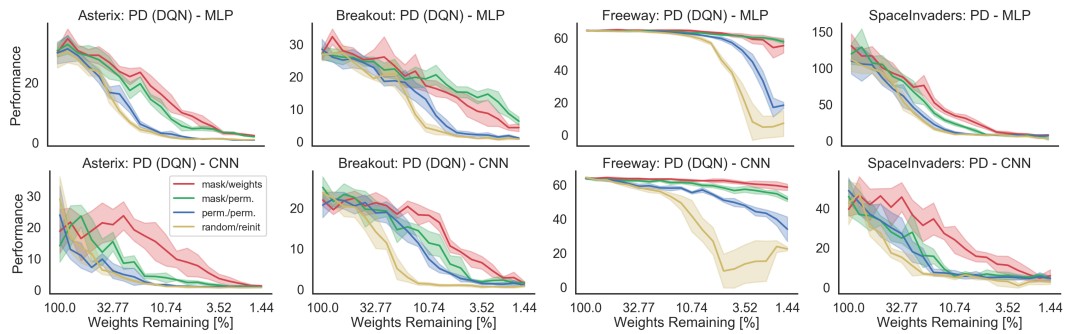

Figure 9: Lottery tickets in supervised policy distillation (MinAtar environments, Young & Tian, 2019). We find evidence for a strong contribution of the IMP-identified mask to the overall ticket effect. The qualitative baseline comparison generalizes from MLP- to CNN-based agents. **Top.** Disentangling ticket baselines for MLP-based value function estimators across four MinAtar games. **Bottom.** Disentangling ticket baselines for CNN-based value function estimators across four MinAtar games. The results are averaged over 5 independent runs for all MinAtar environments. We plot mean best performance and one standard deviation.

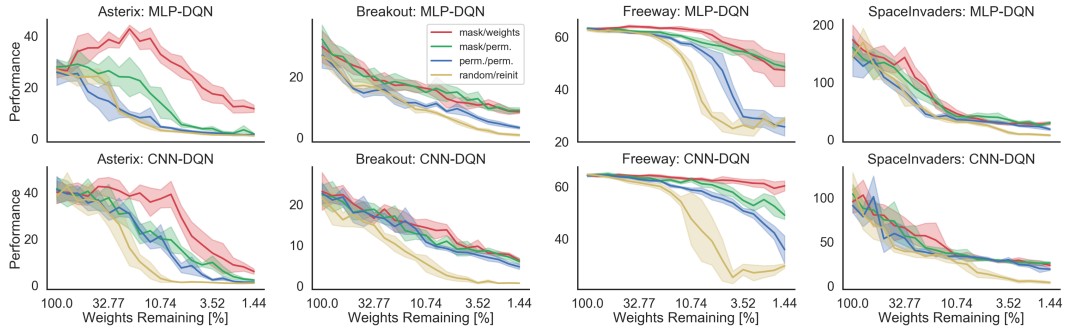

Figure 10: Tickets in off-policy deep reinforcement learning (MinAtar environments, Young & Tian, 2019). We find evidence for a strong contribution of the IMP-identified mask to the overall ticket effect. The qualitative baseline comparison generalizes from MLP- to CNN-based agents. **Top.** Disentangling ticket baselines for MLP-based value function estimators across four MinAtar games. **Bottom.** Disentangling ticket baselines for CNN-based value function estimators across four MinAtar games. The results are averaged over 5 independent runs for all MinAtar environments. We plot mean best performance and one standard deviation.

Strengthening an observation in Yu et al. (2019), we observe that the performance deteriorates at different levels of network sparsity depending on the considered game. Freeway agents keep per-

forming well even for high levels of sparsity, while agents trained on Breakout and Space Invaders continually get worse as the sparsity level increases. In general we find that the qualitative results obtained for MLP agents generalize well to CNN-based agents. The only major difference is that unlike the Asterix CNN agent, the MLP agent improves their performance at moderate levels of sparsity. In summary, we provide further consistent evidence for the strong contribution of the mask to the lottery ticket effect in DRL (both on-policy and off-policy algorithms). The results generalize between different architectures indicating that the strength of the overall lottery ticket effect is mainly dictated by the combination of the task-specification of the environment and the DRL algorithm.

## A.2   THE EFFECT OF NETWORK CAPACITY ON THE ABSOLUTE SIZE OF WINNING TICKETS

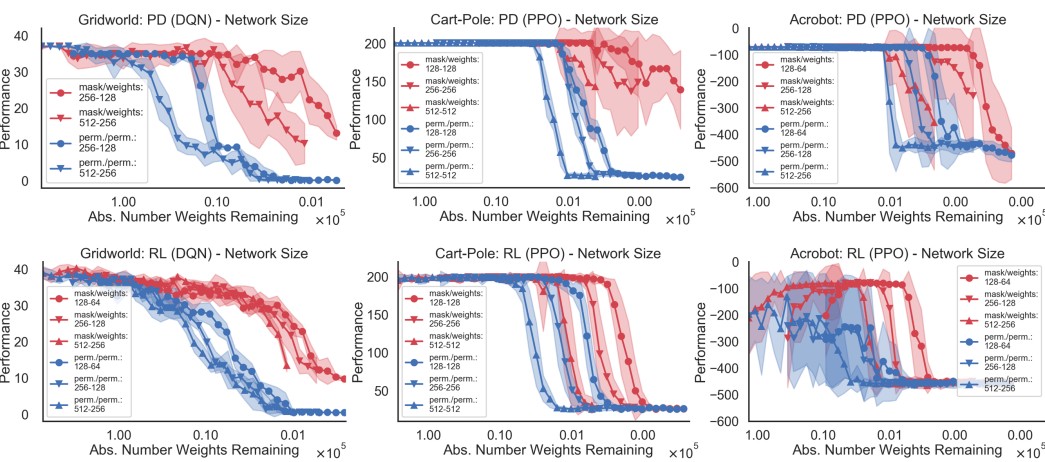

Figure 11: Effect of network size on lottery ticket effect in supervised behavioral cloning and deep reinforcement learning. **Top.** The initial network size has no influence on relative performance of tickets and *permuted/permuted* baselines for supervised behavioral cloning. Larger networks do not yield tickets that outperform those generated from smaller networks for a given absolute number of remaining weights. **Bottom.** Initial network size comparison for tickets in on- and off-policy DRL. Again, larger initial network size does not lead to more performant tickets. The results are averaged over 5 independent runs on the GridMaze environment and 15 independent runs on the Cart-Pole and Acrobot environments. We plot mean best performance and one standard deviation.

The lottery ticket hypothesis suggests that using a larger original network size increases the number of sub-networks which may turn out to be winning tickets (Frankle & Carbin, 2019). To investigate this hypothesis for the case of policy distillation, we analyzed the effect of the initial network size on the lottery ticket effect (figure 2, right column). Against this initial intuition, we observe that smaller dense networks are capable of maintaining strong performance at higher levels of absolute sparsity as compared to their larger counterparts. Furthermore, the initial network size does not have a strong effect on the relative performance gap between the ticket configuration (*mask/weights*) and the baseline (*permuted/permuted*). We suspect that larger networks can not realize their combinatorial potential due to a an unfavorable layer-wise pruning bias introduced by initialization schemes such as the Kaiming family (He et al., 2015). An imbalance between input size and hidden layer size can have strong impact on which weights are targeted by IMP. We further investigate this relationship in section 4.2.

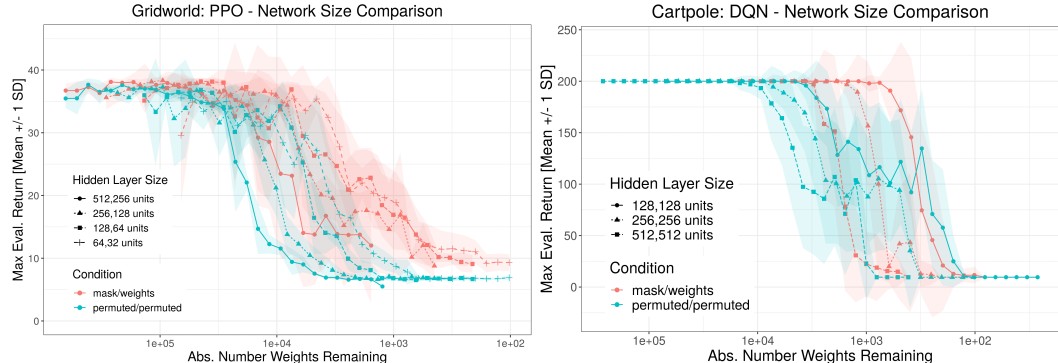

Figure 12: In the main text, agents were trained on GridMaze using the DQN algorithm and on Cart-Pole using PPO. Here, we report the performance of PPO-trained agents on the GridMaze task (left) and of DQN-trained agents on the cart-pole task (right) for different network sizes.

### A.3 THE EFFECT OF DIFFERENT REPRESENTATIONS ON LOTTERY TICKETS IN DRL

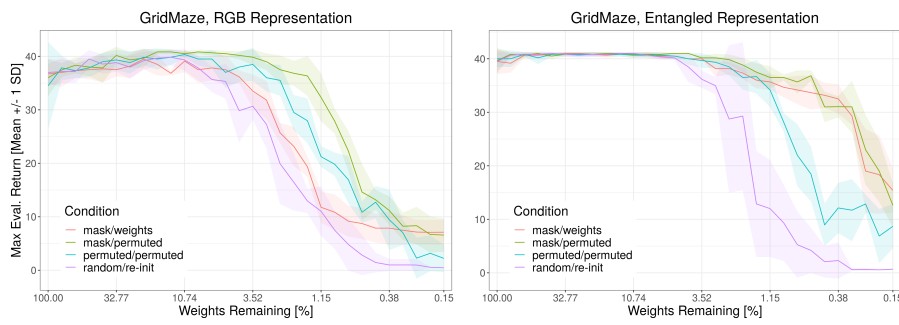

Figure 13: Performance of agents trained on an RGB-encoded GridMaze task (left) and on a randomly projected, entangled representation (right). The derived mask robustly contributes most to the ticket. More information can be found in section B.

### A.4 THE EFFECT OF REGULARIZATION AND LATE REWINDING ON LOTTERY TICKETS IN DRL

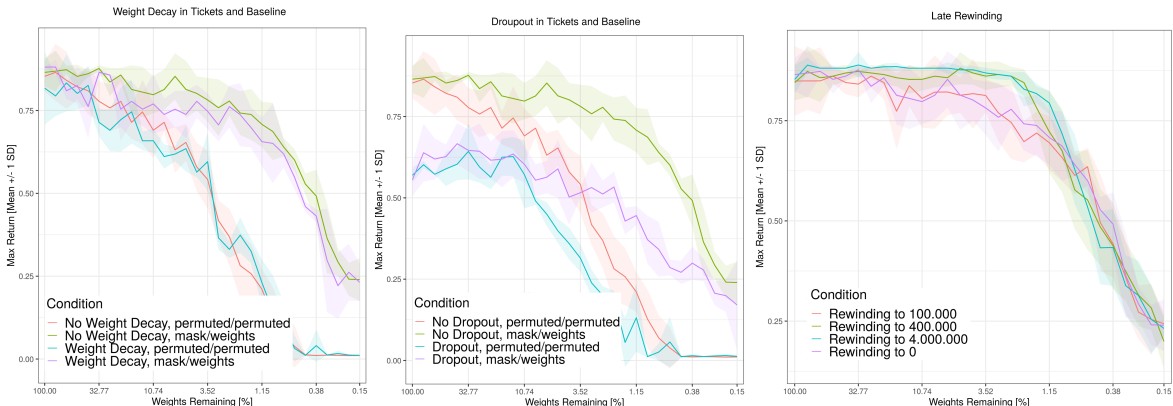

Figure 14: **Left.** Lottery ticket plot with and without L2 weight decay ($\lambda = 0.1$). Using weight decay does not impair the ticket phenomenon for a DQN agent. **Middle.** Lottery ticket plot with and without dropout in all layers ($p = 0.1$). Dropout deteriorates overall performance at all levels of sparsity, but does not impair the ticket effect for a DQN agent. **Right.** Late rewinding (Frankle et al., 2019) to different stages of training (0, 100k, 400k, 4000k environment steps).

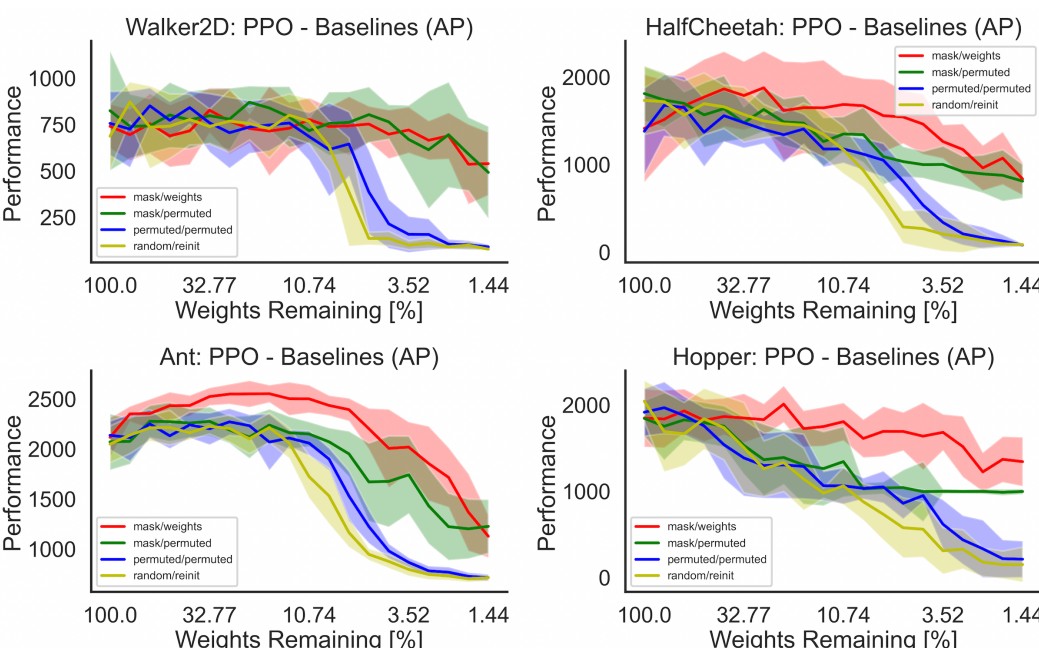

Figure 15: Tickets in on-policy deep reinforcement learning and when pruning only the actor module. The random re-sampling baseline in this case outperforms permuting both the weights and the mask. The results are averaged over 10 runs. We plot mean best performance and one standard deviation.

## B   MAZEGRID ENVIRONMENT

The MazeGrid is a visual navigation task: The agent navigates a grid environment, which is ten pixels high and twenty pixels wide. Each location holds a single unique object. There are six types of objects: empty background (black), walls (grey), the agent (cyan), two moving enemies (magenta and green), as well as 42 coins (yellow) and twelve poisons (brown) (visualized in figure 1, bottom row, left column). The layout of the game is the same in every episode. The agent can walk in four directions (up, down, left, right). The enemies patrol horizontally inside the gaps in the wall. A full game in motion can be watched in the project repository, which will be released after publication. A game terminates after 200 timesteps, or if the agent collects (walks over) all coins, or if the agent is in the same location as an enemy. Each collected coin yields a reward of plus one, each collected poison yields minus one, presented immediately to the agent. To ensure our results do not depend on the specific encoding of the environment, we compared three different representation of the MazeGrid environment (section A.3):

- The **object-map** encoding consists of separate one-hot maps for each type of object. Empty space is encoded explicitly by a separate map, walls however do not have their own map and are only implicitly represented by the lack of any other object. The representation thus contains six maps, ten by twenty pixels each for a total of 1200 binary values for each state.

- The **RGB** encoding consists of the canonical three color channels for each location, resulting in 600 integer values in range $[0, 255]$. Losing information due to occlusion is not an issue in this environment since any location can only hold a single object at a time.

- The **entangled** encoding is derived from the object-map encoding by flattening all values into a vector and multiplying it with a pseudo-random 1200 by 1200 matrix. The matrix's values are sampled independently from $\mathcal{U}(-1, 1)$. The matrix remains constant across all IMP iterations, every seed has its own matrix.

All experiments in the main text were conducted on the object-map environment. Figure 13 provides a comprehensive overview over the baselines to demonstrate that tickets do rely on one specific encoding.The importance of the mask is even further highlighted by our results on RGB specifically.

## C    Hyperparameter Settings for Reproduction

All simulations were implemented in Python using the rlpyt DRL training package (Stooke & Abbeel, 2019, MIT License) and PyTorch pruning utilities (Paszke et al., 2017). The environments were implented by the OpenAI gym (Brockman et al., 2016, MIT License), MinAtar (Young & Tian, 2019, GPL-3.0 License), PyBullet gym (Ellenberger, 2018) packages and the ALE benchmark environment (Bellemare et al., 2013). Furthermore, all visualizations were done using Matplotlib (Hunter, 2007) and Seaborn (Waskom, 2021, BSD-3-Clause License). Finally, the numerical analysis was supported by NumPy (Harris et al., 2020, BSD-3-Clause License). We will release the code after the publication of the paper. The simulations were conducted on a CPU cluster and no GPUs were used. Each individual IMP run required between 8 (Cart-Pole and Acrobot), 10 (MazeGrid, MinAtar) and 20 cores (PyBullet and ATARI environments). Depending on the setting, a for lottery ticket experiment of 20 to 30 iterations lasts between 2 hours (Cart-Pole) and 5 (ATARI games) days of training time.

### C.1    Cart-Pole - Behavioral Cloning & PPO

| Parameter | Value |
|---|---|
| Student Network Size | 128,128 units and 256,256 units |
| Teacher Network Size | 64,64 units and 128,128 units |
| Learning Rate | 0.001 (Adam) |
| Training Environment Steps | 10.000 |
| Number of workers | 4 |
| Distillation Loss | Cross-entropy expert-student policies |

Table 2: Hyperparameters for the **BC** algorithm on **Cart-Pole**. Results reported in fig. 2, 3 and 11.

| Parameter | Value | Parameter | Value |
|---|---|---|---|
| Optimizer | Adam | Value Loss Coeff. | 0.5 |
| Learning Rate | 0.0005 | Entropy Loss Coeff. | 0.001 |
| Temporal Discount Factor | 0.99 | Likelihood Ratio Clip | 0.2 |
| Training Environment Steps | 80.000 | Number of workers | 4 |
| GAE $\lambda$ | 0.8 | Number of epochs | 4 |

Table 3: Hyperparameters for the **PPO** algorithm on **Cart-Pole**. Results reported in fig. 2, 8 and 11.

### C.2    Acrobot - Behavioral Cloning & PPO

| Parameter | Value |
|---|---|
| Student Network Size | 128,64 and 256,128 and 512,256 units |
| Teacher Network Size | 128,64 units |
| Learning Rate | 0.0005 (Adam) |
| Training Environment Steps | 200.000 |
| Number of workers | 4 |
| Distillation Loss | Cross-entropy expert-student policies |

Table 4: Hyperparameters for the **BC** algorithm on **Acrobot**. Results reported in fig. 2, 3 and 11.

| Parameter | Value | | Parameter | Value |
|---|---|---|---|---|
| Optimizer | Adam | | Value Loss Coeff. | 0.5 |
| Learning Rate | 0.0005 | | Entropy Loss Coeff. | 0.01 |
| Temporal Discount Factor | 0.99 | | Likelihood Ratio Clip | 0.2 |
| Training Environment Steps | 500.000 | | Number of workers | 4 |
| GAE $\lambda$ | 0.95 | | Number of epochs | 4 |

Table 5: Hyperparameters for the **PPO** algorithm on **Acrobot**. Results reported in fig. 2, 8 and 11.

### C.3    PYBULLET CONTINUOUS CONTROL - BEHAVIORAL CLONING & PPO

| Parameter | Value |
|---|---|
| Student Network Size | 64,64 Actor and Critic |
| Teacher Network Size | 64,64 units |
| Learning Rate | 0.0005 (Adam) |
| Training Environment Steps | 500.000 |
| Number of workers | 10 |
| Distillation Loss | KL divergence expert-student policies |

Table 6: Hyperparameters for the **BC** algorithm on **PyBullet**. Results reported in fig. 2, and 3.

| Parameter | Value | | Parameter | Value |
|---|---|---|---|---|
| Optimizer | Adam | | Value Loss Coeff. | 0.5 |
| Learning Rate | 0.0005 | | Entropy Loss Coeff. | 0.001 |
| Temporal Discount Factor | 0.99 | | Likelihood Ratio Clip | 0.2 |
| Training Environment Steps | 1.500.000 | | Number of workers | 10 |
| GAE $\lambda$ | 0.98 | | Number of epochs | 10 |

Table 7: Hyperparameters for the **PPO** algorithm on **PyBullet**. Results reported in fig. 4 and 7.

### C.4    GRIDMAZE - DQN

| Parameter | Value | | Parameter | Value |
|---|---|---|---|---|
| Optimizer | Adam | | Replay Buffer Size | 100.000 |
| Learning Rate | 0.0005 | | Replay Buffer $\alpha$ | 0.6 |
| Temporal Discount Factor | 0.99 | | Replay Buffer $\beta$ (init.) | 0.4 |
| Batch Size | 256 | | Replay Buffer $\beta$ (final) | 1 |
| Huber Loss $\delta$ | 1.0 | | Data Replay Ratio | 4 |
| Clip Grad. Norm | 10 | | Training Environment Steps | 5.000.000 |
| $\epsilon^{start}$ | 1 | | $\epsilon^{final}$ | 0.01 |
| #$\epsilon$ Annealing Frames | 1.000.000 | | $\epsilon^{eval}$ | 0.001 |

Table 8: Hyperparameters for the **DQN** algorithm on **GridMaze**. Results reported in fig. 4.

## C.5 ATARI - PPO

| Parameter | Value | | Parameter | Value |
|---|---|---|---|---|
| Optimizer | Adam | | Value Loss Coeff. | 0.5 |
| Learning Rate | 0.0005 | | Entropy Loss Coeff. | 0.001 |
| Temporal Discount Factor | 0.99 | | Likelihood Ratio Clip | 0.2 |
| Training Environment Steps | 2.500.000 | | Number of workers | 10 |
| GAE $\lambda$ | 0.98 | | Number of epochs | 10 |

Table 9: Hyperparameters for the **PPO** algorithm on **ATARI**. Results reported in fig. 8.

## C.6 MINATAR - BEHAVIORAL CLONING & DQN FOR MLP/CNN AGENTS

| Parameter | Value |
|---|---|
| Student Network Size | 1024,512 units (MLP) and Conv2D-16x3, 128-64 (CNN) |
| Teacher Network Size | 512,256 units (MLP) and Conv2D-16x3, 128-64 (CNN) |
| Learning Rate | 0.0005 |
| Training Environment Steps | 10.000.000 (MLP) and 5.000.000 (CNN) |
| Independent Seeds | 5 |
| Distillation Loss | Huber loss of expert-student value estimates |

Table 10: Hyperparameters for the **BC** algorithm on **MinAtar**. Results reported in fig. 9.

| Parameter | Value | | Parameter | Value |
|---|---|---|---|---|
| Optimizer | Adam | | Replay Buffer Size | 100.000 |
| Learning Rate | 0.00025 | | Replay Buffer $\alpha$ | 0.6 |
| Temporal Discount Factor | 0.99 | | Replay Buffer $\beta$ (init.) | 0.4 |
| Batch Size | 256 | | Replay Buffer $\beta$ (final) | 1 |
| Huber Loss $\delta$ | 1.0 | | Data Replay Ratio | 4 |
| Clip Grad. Norm | 10 | | Training Environment Steps | 15.000.00 |
| $\epsilon^{start}$ | 1 | | $\epsilon^{final}$ | 0.05 |
| #$\epsilon$ Annealing Frames | 7.500.000 | | $\epsilon^{eval}$ | 0.001 |

Table 11: Hyperparameters for the **DQN** algorithm on **MinAtar**. Results reported in fig. 10.

