# OpenReview forum: "On Lottery Tickets and Minimal Task Representations in Deep Reinforcement Learning"
_ICLR.cc/2022/Conference — ICLR 2022 Spotlight_

### Official Review · Reviewer_HrAw · 2021-11-01

**Correctness:** 4
**Technical Novelty And Significance:** 2
**Empirical Novelty And Significance:** 3
**Recommendation:** 8
**Confidence:** 3

**Main Review:**

### Strengths

- The principal strength of the paper is in the thoroughness of its empirical evaluations.
    - In particular, the paper identifies a key source of the distinction between RL and supervised learning relevant to lottery tickets: that of distribution shifts induced by the policy improvement operator over the course of training in RL, something absent in supervised learning.
    - The distinction between policy-based and value-based RL is important and I was happy to see both value-based and policy-based agents included in the experimental evaluations.
    - Similarly, I am aware of many papers which study the effect of variations on the IMP masking procedure (e.g. Frankle et al. 2019, 2020; Yu et al., 2020; Zhou et al., 2020), but the clarity and comprehensiveness of the ablation in this work makes it a worthwhile contribution to the literature. I also appreciated that the effect of convolutional layers was discussed in the main paper, and that late rewinding was also evaluated in the appendix.
- The connection between magnitude pruning and state abstraction, particularly in low-dimensional input settings, presents an interpretable and intuitive visualization of lottery tickets.
- The paper is clearly written and it is easy to see how the experimental results justify the claims made in the text.
- While prior works have studied lottery tickets in RL (Yu et al., 2020), this work is distinguished by its focus on the interaction between the non-stationarity inherent in RL and magnitude pruning methods.
- The paper provides useful recommendations for training sparse networks in RL tasks, such as noting the importance of the weight initialization scheme in determining how aggressively inputs are pruned.

### Weaknesses

- The focus of the paper at the moment seems to be “about” the LTH, with RL as an experimental condition in which to study it. In contrast, I think the paper also has the potential to tell us a lot about properties of deep RL agents using the LTH as an experimental condition. The degree to which a network can be pruned seems to be telling us something really interesting about the complexity of a given task, and I think there’s potential to draw a deeper connection to prior work studying non-stationarity in RL.
- The paper dives quite quickly into its findings, but leaves some gaps in the motivation of its investigation. At the moment, my interpretation of the motivational paragraph is that it essentially says “People are interested in lottery tickets but haven’t looked at them properly in deep RL yet, so that is what we will do”. A sentence or two justifying why looking at lottery tickets in RL is interesting and important, beyond simply an absence of existing literature, would greatly benefit the paper. For example, would lottery tickets enable more efficient control systems? Would this investigation tell us something important about deep neural networks?
While it is interesting to look at “retraining” a pruned network from scratch on an RL task, it’s not clear that this is the best approach to sparsify RL agents. As the motivation of IMP is largely to find computationally efficient sparse subnetworks that perform well on the target task, finding a maximally sparse network which attains high return in the target task seems desirable. In this case, presumably a better approach to obtaining sparse agents would be to distill the sparsified network on the outputs of the dense network as a form of supervised training, rather than forcing the network to maintain the capacity necessary to fit all of the intermediate policies and value functions necessary to reach the final high-performing policy.
- I am somewhat skeptical of the interpretation of the convolutional masks in Figure 6. While I can see how each of the labels might correspond to their respective mask, I think it might be reaching a bit to say that particular horizontal lines correspond to bullets as opposed to cannons. I agree with the overall take that magnitude pruning preserves semantically significant components of the filters, but in contrast to the thorough and careful analysis of the rest of the paper, the mapping from the mask on a particular filter into a specific interpretation seems less than rigorous.


### Potential Improvements/Questions
- While behaviour cloning is one way to introduce supervision into RL tasks, I would also be interested in seeing what happens when policy evaluation of an expert policy is used as the supervised task, to complement the comparison between policy-based vs value-based agents trained with policy improvement.
- A priori I would have thought that the value prediction networks would be more sensitive to pruning than policy networks, and the paragraph at the top of page 6 seems to support this intuition. I think it would be easier to parse the paper if it is made clearer in the figures which pruning scheme is used in which evaluations.

**Summary Of The Paper:**

This paper empirically studies the lottery ticket hypothesis in deep reinforcement learning. It confirms the existence of lottery tickets in RL agents, and provides a detailed study of lottery tickets in both control problems and supervised behavioural cloning problems to identify the effect of RL-specific distribution shifts and to evaluate properties of the weight masks produced by iterative magnitude pruning.


**Summary Of The Review:**

Overall, I think the careful analysis and clear presentation of the findings of this paper make it a valuable contribution to both the lottery ticket hypothesis literature as well as to our understanding of non-stationarity in deep RL. I therefore recommend that it be accepted to ICLR.

---

> ### Author Response · Authors · 2021-11-18
> **Reply - Reviewer HrAw**
>
> We thank the reviewer for their detailed suggestions and improvement recommendations.
>
> > "I think the paper also has the potential to tell us a lot about properties of deep RL agents using the LTH as an experimental condition. The degree to which a network can be pruned seems to be telling us something really interesting about the complexity of a given task, and I think there’s potential to draw a deeper connection to prior work studying non-stationarity in RL."
>
> We fully agree that our study provides ample room for further work on non-stationarities in RL. A related point was raised by reviewer BwRq. In a way, our study introduces the lottery ticket procedure as a promising tool for such analyses. However, we fear that an in-depth study is beyond the scope of this revision. We'll be very happy to extend our RL-specific discussion to provide an outlook on how our insights could contribute to understanding the non-stationarities in RL.
>
> > The focus of the paper at the moment seems to be “about” the LTH, with RL as an experimental condition in which to study it.
>
> Yes, the paper is mostly about RL, but we also want to highlight that our proposed baseline investigation and sparse representation analysis also goes beyond the RL setting and can be used to investigate the lottery ticket phenomenon more generally.
>
> > "A sentence or two justifying why looking at lottery tickets in RL is interesting and important, beyond simply an absence of existing literature, would greatly benefit the paper."
>
> Yes, we agree that our motivation could be improved, and thank the reviewer for the recommendation. We will adjust the main text, also taking into account the recommendations of reviewer BwRq regarding the sources of the distribution shift.
>
> > "I am somewhat skeptical of the interpretation of the convolutional masks in Figure 6. While I can see how each of the labels might correspond to their respective mask, I think it might be reaching a bit to say that particular horizontal lines correspond to bullets as opposed to cannons."
>
> We believe that this is an unfortunate misunderstanding. We are not visualizing the upsampled activation maps of a convolutional layer, but the fully connected input layers for a MinAtar environment. In MinAtar, the channels of the observation represent a binary encoding indicating the presence of an object. The labels of the different maps are therefore defined by the environment and not an interpretation by us. We have updated the main text in order to clarify this and excuse the inconvenience.
> But indeed your observation highlights that the presented "visual" interpretation is partially constrained to MLP-based architectures. For convolutional layers the mask would have to prune entire rims of all filters and it would depend both on the stride and padding. For the considered continuous control tasks this is not a problem since most used architectures in the literature are fairly small MLPs. Furthermore, the strong contribution of the mask to the lottery ticket effect can robustly be identified even for convolutional architectures (see ATARI and CNN MinAtar experiments in the supplementary material).
>
> > "While behaviour cloning is one way to introduce supervision into RL tasks, I would also be interested in seeing what happens when policy evaluation of an expert policy is used as the supervised task, to complement the comparison between policy-based vs value-based agents trained with policy improvement."
>
> In the supplementary material (figure 9 and 10) we provide initial evidence for value-based distillation on a set of MinAtar environments. The results for Breakout and Freeway indicate a similar finding for the policy evaluation setting. We did not emphasize these results in the main text, since sometimes we faced the difficulty that some of the distilled agents never trained to full RL performance even at full network capacity. We can only speculate that this may be due to the common observation that one needs a larger student network in order to effectively distill a teacher.
>
> > "A priori I would have thought that the value prediction networks would be more sensitive to pruning than policy networks, and the paragraph at the top of page 6 seems to support this intuition. I think it would be easier to parse the paper if it is made clearer in the figures which pruning scheme is used in which evaluations."
>
> Yes, we agree and have updated the figure titles.

---

> > ### Comment · Reviewer_HrAw · 2021-11-30
> > **Thanks to the authors**
> >
> > Thanks to the authors for their response, in particular on the clarification of Figure 6. This has adequately addressed my concerns, and I remain happy to recommend the paper for acceptance.

---

### Official Review · Reviewer_SmNw · 2021-11-01

**Correctness:** 3
**Technical Novelty And Significance:** 2
**Empirical Novelty And Significance:** 3
**Recommendation:** 5
**Confidence:** 4

**Main Review:**

Strength

1. The paper targets the important problem of dimensionality reduction of task representation which helps in improving performance of several applications where inference response plays a key role.

2. Such empirical studies are extremely important and it provides valuable insights to researchers interested in model optimization in the context of control problems. This study will definitely motivate further work into developing newer classes of sparsification/pruning algorithms in the context of deep RL.


Weakness

1. The intent of the paper is at times difficult to realize .. since the paper has not been structured well. The contributions listed in the introduction does not align well with the arrangement of the sections and respective discussions.

2. Many of the experimental details are pushed to the appendix. This is an important study and researchers will want to try their hands on the the actual experiments and validating their results against the results presented here. So more details on experimental settings .. choice of exp env. ... system reqs etc. is needed.

3. Novelty over state of the art is not clear. While it is understandable that this is an empirical analysis, the reader will still expect at least one of 2 things
       3a. One liner empirical outcome. As in what is right / wrong / broken/ beautiful  about the thing there is being studied
       3b. What can be done about it? A proposal of some sort at least.


**Summary Of The Paper:**

This paper investigates the Lottery Ticket hypothesis in the context of deep RL for identification of sparse task representation in low-dimensional control tasks. This is primarily an empirical investigation where several experiments and consequent analysis reveal what a "winning ticket" means in case of policy models or Q function approximators given that Deep RL is prone to gradual distribution shift. The experiments also help to identify how standard magnitude pruning will behave under different environment updates, feedback schedules or initialization dynamics.

**Summary Of The Review:**

This an important study and analysis that will motivate further research. However the current version has some major concerns that the authors need to address.

---

> ### Author Response · Authors · 2021-11-18
> **Reply - Reviewer SmNw**
>
> We thank the reviewer for their time, feedback and suggestions.
>
> > "The intent of the paper is at times difficult to realize .. since the paper has not been structured well. The contributions listed in the introduction does not align well with the arrangement of the sections and respective discussions."
>
> We agree that the structure of our paper does not follow the "linear" structure of many Machine Learning papers, which try to establish a new state-of-the-art (SOTA), simply because the results do not lend themselves to a linear narrative. In fact, we spent considerable effort trying to compress the results into an accessible format. Of course, we'd be very happy to improve the structure of the text and make it more reader-friendly. Could you maybe elaborate on your point and maybe give us precise pointers to work on?
>
> > "Many of the experimental details are pushed to the appendix. This is an important study and researchers will want to try their hands on the actual experiments and validating their results against the results presented here."
>
> As highlighted by reviewer BwRq, we conducted many experiments using different pruning procedure baselines, environments and RL/BC algorithm settings. Given the page limit it was challenging to fit all of the details into the main text without degrading the overall readability. Therefore, we chose to refer to the supplementary material and provide hyperparameters, environment description and further experiments therein. If you came across specific details that were missing for a full reproducibility, we'd be grateful for hints. Furthermore, we will release the code after the review process.
>
> > "Novelty over state of the art is not clear."
>
> It is not obvious what SOTA refers to in our study. In section 2 (part 3) we contrast our work with the only previous study (Yu et al., 2019) that has started to investigate lottery tickets in the context of RL. Our work extends this study in many aspects: We contrast supervised BC and RL sparse trainability of lottery tickets, introduce baseline conditions to disentangle ticket contributions and analyze the resulting sparse representations. Also, to the best of our knowledge, no previous lottery ticket study has investigated the interpretability of the sparsity masks. We'd be happy to further clarify this in the text.

---

### Official Review · Reviewer_BwRq · 2021-11-01

**Correctness:** 3
**Technical Novelty And Significance:** 3
**Empirical Novelty And Significance:** 4
**Recommendation:** 8
**Confidence:** 4

**Main Review:**

**Quality, Clarity, Soundness, Correctness**

The paper is an impressive empirical study covering a vast range of variations, baselines and ablations. All experiments are repeated multiple times and std-deviations are reported which helps with judging the stat. significance of findings. The necessary background and related work is briefly, but well summarized. Experiments are and findings are well presented - though given the sheer number of experiments and results some parts of the paper appear a bit crammed (but there is no easy solution to this given the limitations of a conference-format paper). Experiments and the conclusions drawn appear to be correct to me - though, sometimes aggregate results like Fig 1 right, bottom do not strictly hold for all experiments that went into the aggregate result (Fig 4 bottom row, where there is no very pronounced difference between the red, green and blue lines). My one criticism is the hypothesis that attributes distributional shift due to exploration as the reason for why pruning rations (without performance degradation) in RL are typically lower than in supervised learning. While the hypothesis is sensible, it is never formally verified, and no alternative explanations are presented (one simple control could be to train the supervised baseline on data that includes the various stages of the RL agent, including early exploration phases, rather than just using “expert” trajectories).

**Verdict**

Overall I think this is a great empirical paper. Experiments are plentiful, well executed and with an impressive amount of relevant baselines and controls. While the lottery ticket effect has been investigated in great detail in non-RL domains before, and many ideas and inspiration for experiments could be transferred from previous work, I do not think this limits the merit of the current paper. Important empirical verification was carried out and results were presented in detail, and with some interesting observations. The paper lays the groundwork for now digging deeper into some of the findings. I have some small suggestions for improvements, but am currently in favor of accepting the paper. Looking forward to hearing the other reviewer’s opinion.

**Improvements**

1) As stated earlier, my main criticism is that the “distributional shift” hypothesis is never verified and no alternative explanations are provided as to why pruning rates in RL seem to be generally lower. At the current stage I would phrase the finding as a (sensible) hypothesis that needs further investigation rather than an established finding.
    * 1a: One control experiment that comes to my mind is to run the behavioral cloning baseline on data across the full “lifetime” of an RL agent, including the early exploration phase and thus exhibiting the same distributional shift throughout supervised training.
    * 1b: One alternative explanation not related to distributional shift is that the supervised and RL learning signal might be qualitatively different and thus lead to the different pruning rates (not entirely sure how to test that though).
    * 1c: If the hypothesis is true there should also be observable qualitative effects for tasks where this distributional shift is large compared to tasks where the distributional shift is small.
    * 1d: See 3 for more alternatives related to exploration.

2) Another limitation that should ideally be noted more explicitly is that most findings are based on PPO (an on-policy algorithm) with fewer experiments for DQN (off-policy). Large parts of the paper are written as if the findings were established in general about RL. While the paper certainly does a great job of supporting the main claims with broad sets of experiments, I think it would be nice to point out a small hint of caution (e.g. in the limitations section) regarding the generality of findings in the off-policy setting.

3) Clarification: how is exploration in the RL algorithms implemented? Is it baked in via a fixed epsilon-greedy action-selection? Is there some annealing of the exploration strength?
    * 3a: If the policy has to account for exploration as well (e.g. through some entropy regularization) this might be another explanation why pruning rates in RL are lower - because the policy needs to be more complex than the supervised policy.
    * 3b: If the RL algorithm is using fixed epsilon-greedy (no annealing), is this taken into account when generating the expert data for the behavioral cloning baseline (i.e. using an expert with the same epsilon)? If not, the data-distribution for RL (even at the end of training) is potentially much broader compared to the behavioral cloning baseline which could also be a reason for the lower pruning rates in RL.


**Summary Of The Paper:**

**Update after reading the other reviews and authors' responses:** Some valid criticism has been raised and addressed by the authors to a reasonable degree (given e.g. the limitations of a conference-format paper). Given all current information I remain in favor of accepting the paper (my score and confidence remains unchanged).

**Summary:**

The paper investigates the lottery-ticket effect in detail in reinforcement learning tasks. While it has been shown before that lottery-tickets also exist in deep reinforcement learning, this paper performs a thorough analysis using (i) both discrete and continuous action- and observation-spaces, (ii) an on- and an off-policy RL algorithm (as well as a behavioral cloning baseline), and (iii) careful ablations to distinguish the importance of the binary mask and the initial weight values which together form a winning ticket. Various controls and ablations are performed, leading to a plethora of experimental results. The main result is that winning tickets can be found efficiently via iterative magnitude-based pruning in RL tasks, and that in many tasks the binary mask contributes more to performance than the initial weight values. Compared to the supervised (behavioral cloning) baseline, RL performance degrades more rapidly with increasing sparsity levels which is attributed to the distributional shift introduced by exploration in RL. Finally, the paper also investigates the effect of pruning (in winning tickets) of the first-layer weights, and shows that in many cases multiple input dimensions (or regions, which are often semantically meaningful) can be pruned without loss in task performance. This observation could be an interesting approach to producing sparse input representations in RL.

**Main contributions**

1) Investigation of the lottery ticket effect in RL. Significance: though it has been reported before that winning tickets can be found in RL, the paper performs an impressive amount of empirical investigations, including many baselines and ablations across a range of continuous and discrete tasks and an on- and off-policy RL algorithm as well as a supervised baseline. Naturally, results vary somewhat between settings and variations, but the sheer number of results allows to extract trends that hold across many variations. This significantly increases the reliability of the findings.
2) Disentangling the lottery ticket effect through control experiments that allow to attribute the effect strength to (i) the binary mask, (ii) the initial weight-values, or (iii) the layer-specific pruning ratios, which are all combined in a winning ticket. Significance: these control experiments shed some important light on the role of the three components. Interestingly, the binary mask seems to play the most important role in RL tasks, whereas initial weight-values seem to matter more in the supervised setting. This is an interesting, and very consistent insight which might spawn further investigation into the topological structure induced by the winning mask.

3) Investigation of first-layer weights of winning tickets. In many cases whole input dimensions (in continuous tasks) or input regions (in discrete tasks) that are irrelevant to the problem are pruned. This reflects an important (implicit?) regularizing effect - the corresponding weights remain at low magnitudes and can be safely pruned. Investigation of the pruning patterns or pruned dimensions also provides semantic insight which allows for interesting hypotheses regarding how the network solves the task and which information it ignores. Significance: this is an interesting artifact of the analysis of winning tickets in RL. Results are promising and ask for further investigation and perhaps even formulation of a method for sparse/compressed input representations for RL (the identified irrelevant dimensions also transfer to non-pruned architectures).


**Summary Of The Review:**

An impressive empirical paper that investigates many findings regarding the lottery ticket effect found in other domains to RL tasks and algorithms. The paper presents results for a large number of tasks (with continuous and discrete observation- and action-spaces), and loads of relevant controls, baselines and ablations. The main finding is that iterative magnitude-based pruning can effectively find winning tickets in RL, and that there are some qualitative differences between RL and supervised learning (masks seem to be more relevant than initial-weight values of winning tickets). I only have some small criticism regarding one hypothesis, and whether it can be stated as empirically verified given the current set of experiments. Currently I think the paper is interesting and relevant to a large part of the ICLR community and vote in favor of accepting the paper.

---

> ### Author Response · Authors · 2021-11-18
> **Reply - Reviewer BwRq**
>
> We thank the reviewer for the thoughtful comments and would like to take the opportunity to discuss some of the suggestions.
>
> > "[..] my main criticism is that the “distributional shift” hypothesis is never verified and no alternative explanations are provided as to why pruning rates in RL seem to be generally lower."
>
> Thank you, we fully agree. It would have been great to properly prove this hypothesis, but we found this rather non-trivial, as detailed below. We do have more concrete evidence, but up to now we believe it's too anecdotal for publication.
>
> > "1a: One control experiment that comes to my mind is to run the behavioral cloning baseline on data across the full “lifetime” of an RL agent, including the early exploration phase and thus exhibiting the same distributional shift throughout supervised training."
>
> That's a great suggestion, but we feel that it is also a slightly unfair comparison, because the network would face the more difficult task of simultaneously learning the exploration and the exploitation behaviour, which would all be treated as validly labelled, although potentially conflicting. Therefore, an earlier decline of the performance-pruning curve would be somewhat inconclusive, because it could merely arise from the increased task complexity. In a way, your suggestion sounds like the Offline RL protocol (e.g. Levine et al., 2020). We are not entirely sure though whether or not agents would still be capable of learning without using dedicated Offline RL algorithms in this case (see e.g. figure 1 of Kumar et al., 2019 for the SAC case). How could we compose the data buffer in a "fair" manner? How much data from the exploration and how much from the exploitation phase? We'd probably end up with another set of experimental hyperparameters.
>
> > "1b: One alternative explanation not related to distributional shift is that the supervised and RL learning signal might be qualitatively different and thus lead to the different pruning rates (not entirely sure how to test that though)."
>
> Our empirical findings agree with your assessment regarding the quality of the credit assignment signal. In general, BC agents can be trained using far less step transitions. Throughout our experiments we therefore increased the amount of training steps for RL agents to ensure that in principle all agents are receiving enough experience to learn even at high degrees of sparsity. Still BC agents can consistently be trained at higher levels of sparsity.
>
> > "1c: If the hypothesis is true there should also be observable qualitative effects for tasks where this distributional shift is large compared to tasks where the distributional shift is small."
>
> Yes, that is also what we expected, but at this point, we only have anecdotal evidence. For example, we initially considered smaller gridworlds in which even the converged agent had to visit all states, such that the distributional shift should be considerably smaller. In this case, we did not find a clear difference between the sparse trainability of RL and BC tickets. Hence, we agree that there seems to be a degree of task dependence. Unfortunately, it is hard to control the amount of expected distribution shift by handpicking environments, let alone quantify it. This makes it practically difficult to test your hypothesis thoroughly, so we're afraid that this is beyond the scope of this revision. If you have concrete ideas, we'd of course be very happy to check.
>
> > Re: Note of caution regarding that most results stem from On-Policy PPO and not Off-Policy DQN.
>
> Yes, we agree and appreciate the detailed observation. We added a paragraph to the limitations section of the paper.
>
> > "Clarification: how is exploration in the RL algorithms implemented? Is it baked in via a fixed epsilon-greedy action-selection? Is there some annealing of the exploration strength?"
>
> Our DQN experiments use epsilon-greedy exploration with linear epsilon annealing from 1 to 0.01 within 20 percent of the training frames. Evaluation is performed with epsilon being 0.001. We added this to the supplementary material section C4. The PPO agents, on the other hand, are indeed trained with a small amount of entropy regularization (coefficient of 0.001, no annealing).
>
> > 3b: If the RL algorithm is using fixed epsilon-greedy (no annealing), is this taken into account when generating the expert data for the behavioral cloning baseline (i.e. using an expert with the same epsilon)?
>
> The value-based distillation results we discuss in the supplementary material (figures 9 and 10) all are based on a student value prediction network, which is trained on the values predicted by an expert teacher, which is rolled out with an epsilon of 0.001 (i.e., the evaluation epsilon of the RL agent), so we believe that there are no risks here.

---

> > ### Comment · Reviewer_BwRq · 2021-11-20
> > **Thanks for the detailed response and clarification**
> >
> > I want to thank the authors for the detailed response. I agree with their responses and only have some minor further comments (below). I am happy that the authors managed to address some of the potential improvements / issues pointed out by the reviewers (e.g. slight improvements to the motivation). I want to re-iterate my recommendation to phrase the "increased distributional shift due to exploration in RL" as the cause for lower sparsity-rates as a hypothesis (with some empirical evidence in the paper pointing towards the hypothesis) that requires further investigation, rather than a fully established finding. Ideally this is accompanied by a brief discussion on alternative hypotheses that cannot be ruled out currently.
> >
> > Having read the other reviews and the authors' response I remain in favor of accepting the paper and my score remains unchanged. I think the feedback/criticism raised by SmNw is valid (and should be addressed as well as possible in the intro, discussion and conclusion) - however, I personally do not see an easy way to address missing experimental details in the main paper (given the limitations of a conference format and the number of experiments conducted), and also don't see an obvious way to "compress" the paper into a single proposal/improvement/outcome given the investigative nature of the work (as opposed to, say, a novel-method paper).
> >
> > Comments re. authors' response:
> > 1a: I agree with the difficulties of composing the data buffer in a fair manner and have no canonical solution to offer. Regarding the "fairness" due to the increased complexity: I agree with the authors' points, but perhaps it is exactly this increased complexity of the policy in RL that causes the lower pruning rates - that's also why the question of how exploration is implemented in RL is important (e.g. for an entropy-regularized policy, the entropy-regularizer prevents the policy from going below a certain complexity, which should lie above the complexity of a policy distilled from a epsilon-greedy expert with low epsilon; on the other hand an agent trained via epsilon-greedy exploration potentially needs a less complex policy since the "exploration" is implemented "outside the policy" via the exploration noise). Long story short: the comparison shown to BC in the paper is interesting and meaningful; in general it would be nice to have controls/ablations where the following two factors can be disentangled:
> >  * Increased complexity (and thus lower sparsity rates) of the RL policies which have to capture both exploratory and exploitative behavior (whereas the supervised policy only needs to capture exploitative behavior).
> >  * Increased distributional shift and its relation to sparsity rates (e.g. through finding tasks with different "amounts" of distr. shift).
> > Typically both factors are coupled, and I think it is beyond the scope of this paper to disentangle both (the latter is important for verifying that the "distributional shift hypothesis", but to me the main point of the current paper is to establish qualitative and quantitative properties of lottery tickets in RL). So while it would be nice to have such experiments, I think they are quite hard to design and conduct in a meaningful way, which probably deserves a separate publication.
> >
> > 1b, 1c: Thanks for the comments and anecdotal results - I fully agree that doing a thorough investigation of these points is beyond the scope of the paper, and is not needed for the validity of the paper's main finding.
> >
> > 3b: Thanks for the additional details regarding the implementation of exploration - I agree that using the same epsilon for the teacher that is used to evaluate the RL agent is a very sensible choice that should make results as comparable as possible.

---

### Decision · Program_Chairs · 2022-01-20

**Decision:**

Accept (Spotlight)

**Comment:**

This paper studies the Lottery Ticket hypothesis in reinforcement learning for identifying good sparse representations for low-dimensional tasks. The paper received initial reviews tended towards acceptance. However, the reviewers had some clarification questions and concerns. The authors provided a thoughtful rebuttal. The paper was discussed and most reviewers updated their reviews in the post-rebuttal phase. Reviewers generally agree that the paper should be accepted but still have good feedback. AC agrees with the reviewers and suggests acceptance. However, the authors are urged to look at reviewers' feedback and incorporate their comments in the camera-ready.